# Diversity and Biocontrol Potential of Cultivable Endophytic Bacteria Associated with Halophytes from the West Aral Sea Basin

**DOI:** 10.3390/microorganisms9071448

**Published:** 2021-07-06

**Authors:** Lei Gao, Jinbiao Ma, Yonghong Liu, Yin Huang, Osama Abdalla Abdelshafy Mohamad, Hongchen Jiang, Dilfuza Egamberdieva, Wenjun Li, Li Li

**Affiliations:** 1State Key Laboratory of Desert and Oasis Ecology, Xinjiang Institute of Ecology and Geography, Chinese Academy of Sciences, Urumqi 830011, China; gaolei19@mails.ucas.ac.cn (L.G.); majinbiao@ms.xjb.ac.cn (J.M.); liuyh@ms.xjb.ac.cn (Y.L.); yinhuang0929@163.com (Y.H.); osama@aru.edu.eg (O.A.A.M.); jiangh@cug.edu.cn (H.J.); 2College of Resources and Environment, University of Chinese Academy of Sciences, Beijing 100049, China; 3State Key Laboratory of Biogeology and Environmental Geology, China University of Geosciences, Wuhan 430074, China; 4Faculty of Biology, National University of Uzbekistan, Tashkent 100174, Uzbekistan; egamberdieva@yahoo.com; 5Leibniz Centre for Agricultural Landscape Research (ZALF), 15374 Müncheberg, Germany; 6State Key Laboratory of Biocontrol, Guangdong Provincial Key Laboratory of Plant Resources, School of Life Sciences, Sun Yat-Sen University, Guangzhou 510275, China

**Keywords:** endophytes, halophytes, diversity, antifungal activity, Aral Sea

## Abstract

Endophytes associated with halophytes may contribute to the host’s adaptation to adverse environmental conditions through improving their stress tolerance and protecting them from various soil-borne pathogens. In this study, the diversity and antifungal activity of endophytic bacteria associated with halophytic samples growing on the shore of the western Aral Sea in Uzbekistan were investigated. The endophytic bacteria were isolated from the nine halophytic samples by using the culture-dependent method and identified according to their 16S rRNA gene sequences. The screening of endophytic bacterial isolates with the ability to inhibit pathogenic fungi was completed by the plate confrontation method. A total of 289 endophytic bacterial isolates were isolated from the nine halophytes, and they belong to *Firmicutes*, *Actinobacteria*, and *Proteobacteria*. The predominant genera of the isolated endophytic bacteria were *Bacillus*, *Staphylococcus*, and *Streptomyces*, accounting for 38.5%, 24.7%, and 12.5% of the total number of isolates, respectively. The comparative analysis indicated that the isolation effect was better for the sample S8, with the highest diversity and richness indices. The diversity index of the sample S7 was the lowest, while the richness index of samples S5 and S6 was the lowest. By comparing the isolation effect of 12 different media, it was found that the M7 medium had the best performance for isolating endophytic bacteria associated with halophytes in the western Aral Sea Basin. In addition, the results showed that only a few isolates have the ability to produce ex-enzymes, and eight and four endophytic bacterial isolates exhibited significant inhibition to the growth of *Valsa mali* and *Verticillium dahlia*, respectively. The results of this study indicated that halophytes are an important source for the selection of microbes that may protect plant from soil-borne pathogens.

## 1. Introduction

In the past about 50 years, due to the frequent interruption of human activities and the increasing irrigation water of agriculture, the water area of the Aral Sea has been decreasing, and the salinity of the Aral sea has been rising, which has given rise to a series of ecological and environmental problems [1,2,3]. The soil salinization is problematic in many parts of the world and causes severe food crises through reducing agricultural lands for crop production [4]. Previous reports indicated that salinity increases the susceptibility of plants towards various phytopathogens and is considered as a major factor affecting crop growth and cereal yield [5,6,7,8,9]. Since chemical control of plant diseases has negative effects on environment [10], biological control agents based on microbes is one of the alternative ecological safe approaches to protect plants from pathogens [11,12].

Halophytes have evolved several strategies to withstand abiotic and biotic stresses through their associated microbiome [4,13]. Endophytes are a group of organisms (such as bacteria or fungi) living within a plant, being found in almost all plants worldwide [14]. Endophytes play a very vital role in the improvement of plant growth and development, the phytoremediation of contaminated environments by production of a variety of secondary metabolites with special functional activities, and the plant resistance to plant diseases caused by pathogenic fungi [15,16,17,18]. Therefore studying the endophytic bacterial diversity and their antifungal activity is of great importance in developing biological control agents. In one review related to *Streptomyces* as efficient colonizers of plant tissues, the author summarized that metabolites like antibiotics and volatile organic compounds may have great potential as excellent agents for controlling various fungal and bacterial phytopathogens [19]. You et al. [20] isolated one endophytic bacteria isolate BZJN1, identified as *Bacillus subtilis* by physiological experiments, biochemical tests, and molecular identification, and found it can significantly inhibit the growth of *Ceratobasidium* sp., a group of plant pathogenic fungi. Wang et al. isolated a total of 165 cultivable endophytic bacterial isolates from the *Dendrobium*, a traditional medicinal plant in China, and 14 endophyte bacterial isolates able to control phytopathogen were detected using the Kirby-Bauer method [21]. Experiments such as that conducted by Bibi (2018) have shown that the endophytic bacteria from seagrass have the characteristics of diversity and wide antagonistic effects on phytopathogen [22].

In this study, we reported the diversity of cultivable endophytic bacteria associated with nine halophytes collected from the western Aral Sea Basin in Uzbekistan and their antifungal activity. The objectives of our study were: (1) to isolate and identify cultivable endophytic bacteria associated with nine halophytes growing in saline soil by using the culture-dependent method and 16S rRNA gene sequencing; (2) to determine the best medium for isolating endophytic bacteria associated with halophytes from the western Aral Sea Basin; (3) to explore the differences of cultivable endophytic bacteria isolated from halophytes in different exposed zones of the western Aral Sea Basin; and (4) to evaluate the antifungal activities of the isolated endophytic bacterial isolates.

## 2. Materials and Methods

### 2.1. Sample Collection and Site Description

According to the follow-up survey records of the Institute of Natural Sciences, Academy of Karakalpakstan, from the 1970s to 2018, owing to the shrinking water area, the western Aral Sea has formed a huge exposed zone, which extended to about 1500 m from the shoreline to the far shore [23]. We defined the original exposed zone as E1970S, followed by E1980S, E1990S, E2000S, and E2009S, which indicated that these soil bands were exposed in about the 1970s, 1980s, 1990s, 2000s, and 2009 respectively. Furthermore, we also collected halophytic samples from ODZ, which means the non-exposed zone or the outside of the degradation zone, to explore the endophytic bacterial diversity completely.

Nine halophyte samples labeled from S1 to S9 respectively were collected in 2018 from the western Aral Sea in Uzbekistan and placed in aseptic bags, which placed ice straightway and transported back to our lab (Table 1). These nine halophyte samples belong to six halophytic species, which are widely distributed on the shore of the Western Aral Sea. This indicated that these halophytes are the dominant species in the local area. Therefore, these dominant plants growing on the shore of the Western Aral Sea were selected to investigate endophytic bacterial diversity and community structure (Appendix A and Table 1).

### 2.2. Sterilization of Plant Materials

All the collected plants were washed under running tap water to remove the soil attached to the root and dust on the plant surface. The plants were sterilized with 75% ethanol for 1 min and subsequently with 5% NaClO for 8 min and were rinsed five times in sterile distilled water [24]. To check the sterility of the surface of plants after surface sterilization of plant materials, we spread 100 μL of the last rinse ddH_2_O on the TSA plate, and found the absence of any colonies after 3-day incubation, which confirmed that sterilization was thorough. The sterile plants were cut into 1–2 cm pieces with a sterile scalpel and dried in the horizontal flow clean bench. Finally, all of the samples were crushed by a sterile masher and stored at −20 °C.

### 2.3. Isolation of Endophytic Bacteria

The plant sample (1 g) was ground using a sterile mortar and transferred into a conical flask with 9 mL of sterile water and shaken for 100 rpm, 30 min using a Shaken Incubator (Shanghai Tensuc Lab Instruments Manufacturing Co., Ltd., Shanghai, China). About 100 μL dilutions (10^−2^–10^−4^) were spread on the isolated plates (Table 2) and the plates were incubated for 15 days at 30 °C. After 15 days, colonies with different shapes and colors were picked and carefully transferred by streaking on Tryptic Soy Agar plates (TSA) and incubated for the next 3 days to check the purity of the isolates. Visually homological colonies in sizes, shapes, and colors were removed from duplicate isolates and reduced the number of isolates to be sequenced.

### 2.4. DNA Extraction, PCR Amplification and 16S rRNA Gene Sequencing

The chelex-100 method was used for DNA extraction. The small parts of the colonies were transferred into PCR tubes with 200 μL 5% chelex-100 and were mixed with a Biosan B-1 Vortex for 10 s. The tubes were incubated at 99 °C for 30 min in a PCR instrument (C1000 Touch™ Thermal Cycler, BioRad Laboratories, Hercules, CA, USA) and centrifuged at 12,000× *g* for 10 min. DNA-containing supernatant was stored at −20 °C for polymerase chain reaction (PCR). The 16S rRNA genes were amplified via PCR using the following primers: 27F (5′-GAGTTTGATCCTGGCTCAG-3′) and 1492R (5′-GAAAGGAGGTGATCCAGCC-3′) (Biomed, Beijing, China) [25]. Extracted DNA was used as a template for 16S rRNA gene amplification. Each 50 µL of reaction mixture contained 4 µL (20–30 ng) template DNA, 25 µL 2×Taq MasterMix (Coolaber, Beijing, China), 2 µL 27F, 2 µL 1492R, and 17 µL DNase/RNase-Free Deionized Water (TIANGEN BIOTECH (BEIJING) CO., Ltd., Beijing, China). The PCR was performed using a C1000 Touch™ Thermal Cycler (Bio-Rad Laboratories, USA). The PCR program was as follows: pre-denaturation at 94 °C for 4 min, followed by 35 cycles of denaturation for 30 s at 94 °C, annealing for 30 s at 55 °C and extension for 1.5 min at 72 °C, followed by the final step for 7 min at 72 °C. The PCR-amplified products were examined by electrophoresis in a 1.2% agarose gel containing GoldviewTM. The PCR products were purified and sequenced in Sangon Biotech, Shanghai, China. Received merged data were compared with those registered in the EZBioCloud database (https://www.ezbiocloud.net/, accessed on 11 December 2020) using a basic online alignment search tool (16S-based ID).

### 2.5. Accession Numbers

The 16S rRNA gene sequences of the endophytic bacteria of nine halophytes were deposited into GenBank under the accession numbers: MW664036–MW664324.

### 2.6. Screening of Bacterial Isolates Producing Protease, Lipase, and Cellulose

Testing the protease activity was determined by growing the isolates on Skim Milk Agar medium. After incubation for 7 days at 30 °C, the positive results were indicated by the formation of a clear halo zone around the bacterial colonies [26]. Cellulose activity was tested by inoculating bacterial isolates onto a medium containing a unique cellulose substrate namely carboxy methyl cellulose sodium salt (CMC-Na) [27]. After 7 days of incubation at 30 °C, all plates were stained with 5 mL 0.1% Congo red solution for 10 min and then rinsed by using 5 mL 1 M NaCl. Positive results were indicated by the formation of a clear or lightly colored zone around the colony. The assay of lipase activity was carried out by using bacterial isolates being scratched on Tween80 medium [28]. The content of Tween80 medium was as follows: 10 mL/L Tween80, 50 mL/L Victoria Blue-suspension, 10 g/L peptone, 5 g/L NaCl, 0.1 g/L CaCl_2_·7H_2_O, and 15 g/L agar. Bacteria isolates were incubated for 7 days at 30 °C and the formation of a clear halo was observed indicating the result is positive. The enzyme production capacity was calculated by the formula (three replicates to get the average enzyme production capacity) [29]. E refers to the enzyme production capacity of endophytic bacteria. D1 refers to the diameter of the clear zone and D2 refers to the diameter of the endophytic bacterial isolate.
(1)$$E=\frac{\;D1}{D2}\times 100$$

### 2.7. Screening of Endophytic Bacterial Isolates for Their Antifungal Activity

The plate confrontation method was used to screen endophytic bacterial isolates with the ability to inhibit pathogenic fungi. In the experimental group, the pathogenic fungi (0.5 cm in diameter) were inoculated in the center of the PDA plate and tested endophytic bacteria were inoculated 2.5 cm from the center with pathogenic fungi. The plate inoculated with pathogenic fungi and not inoculated with tested endophytic bacteria was regarded as a control group. All of the plates were put in an incubator at 25 °C for 7 days. After the pathogenic fungi of CK treatment grew all over the petri dish, the growth diameter of the pathogen was measured and the inhibition rate of endophytic bacteria reisolating the pathogen mycelium was calculated according to the formula (three replicates to get the average inhibition rate) [30]. CK refers to the colony diameter of pathogenic fungi in the control group. EXP refers to the colony diameter of pathogenic fungi in the experimental group; 0.5 cm refers to the colony diameter of pathogenic fungi at the beginning of the inoculation. The pathogens for the antagonistic testing are *Valsa mali*, *Verticillium dahliae*, *Fusarium oxysporum*, and *Fusarium solani*.
(2)$$\mathrm{Inhibition}\;\mathrm{rate}=\frac{\;[(\mathrm{CK}-0.5\;\mathrm{cm})-(\mathrm{EXP}-0.5\;\mathrm{cm})]}{\mathrm{CK}-0.5\;\mathrm{cm}}\times 100$$

### 2.8. Statistical Analysis

Statistical analyses and visualization of all isolation data were completed by Microsoft Excel 2019 and R (version 4.0.3). The flower diagram was produced with the GENESCLOUD, an online drawing tool for scientific research. The Upset plot was created on the website (https://www.omicstudio.cn/tool/43, accessed on 15 December 2020), a web-based drawing tool for scientific research. Furthermore, we use the LSD test of R software to examine the significant difference in inhibition rate among endophytic bacterial isolates.

## 3. Results

### 3.1. Isolation and Identification of Cultivable Endophytic Bacteria

A total of 289 endophytic bacteria belonging to 3 phyla, 6 classes, 14 orders, 17 families, and 19 genera were isolated from nine samples (Figure 1A–C; Appendix A). The isolation results showed that the predominant genera were *Bacillus*, *Staphylococcus*, and *Streptomyces*, accounting for 38.5%, 24.7%, and 12.5% of the total number of isolates, respectively (Figure 1C).

About 26 isolates were isolated and purified from sample S1, belonging to two phyla, two classes, four orders, five families, and five genera. Thirty isolates were isolated and purified from sample S2, belonging to two phyla, two classes, two orders, three families, and three genera. Forty-six isolates were isolated and purified from sample S3, belonging to three phyla, three classes, three orders, four families, and four genera. Forty-nine isolates were isolated and purified from sample S4, belonging to two phyla, two classes, four orders, five families, and five genera. Sixteen isolates were isolated and purified from sample S5, belonging to three phyla, three classes, three orders, three families, and four genera. Sixteen isolates were isolated and purified from sample S6, belonging to two phyla, two classes, three orders, four families, and four genera. Forty-two isolates were isolated and purified from sample S7, belonging to three phyla, four classes, seven orders, seven families, and seven genera. Forty-six isolates were isolated and purified from sample S8, belonging to 3 phyla, 5 classes, 9 orders, 9 families, and 10 genera. Eighteen isolates were isolated and purified from sample S9, belonging to three phyla, three classes, five orders, five families, and five genera. The Shannon diversity indices from sample S1 to S9 figured by the R package vegan are 2.08, 1.44, 1.13, 1.40, 1.24, 1.19, 0.85, 2.52, and 1.61 respectively. The Simpson diversity indices from sample S1 to S9 figured by the R package vegan are 0.79, 0.68, 0.59, 0.73, 0.63, 0.61, 0.34, 0.90, and 0.75 respectively. The Richness indices from sample S1 to S9 counted by the R package vegan are 13.00, 7.00, 7.00, 5.00, 5.00, 5.00, 8.00, 16.00, and 7.00 respectively. Pielou evenness indices from sample S1 to S9 calculated by the R package vegan are 0.81, 0.74, 0.58, 0.87, 0.77, 0.74, 0.41, 0.91, and 0.83 respectively. The statistics of isolation results clearly showed that the diversity and richness of S8 are the highest. The diversity of S7 is the lowest, while the richness of S5 and S6 are the lowest (Table 3).

In the isolation experiments, we found no shared species between the nine samples. The exclusive species of sample S1 include seven species, identified as *Bacillus licheniformis*, *Bacillus siamensis*, *Bacillus subtilis*, *Bacillus tequilensis*, *Micrococcus terreus*, *Streptomyces althioticus*, and *Streptomyces calvus* respectively. The exclusive species of sample S2 include two species, identified as *Bacillus pumilus* and *Streptomyces griseoviridis* respectively. The exclusive species of sample S3 include two species, identified as *Pseudomonas oryzihabitans* and *Streptomyces caeruleatus* respectively. The exclusive species of sample S4 include three species, identified as *Brevibacterium sediminis*, *Corynebacterium glyciniphilum*, and *Streptomyces puniceus* respectively. The exclusive species of sample S5 include three species, identified as *Mesobacillus campisalis*, *Nocardiopsis dassonvillei* subsp. *Albirubida*, and *Novosphingobium lindaniclasticum* respectively. The exclusive species of sample S6 include two species, identified as *Nocardiopsis trehalosi* and *Streptomyces setonii* respectively. The exclusive species of sample S7 include four species, identified as *Clavibacter michiganensis* subsp. *Phaseoli*, *Mycolicibacterium arabiense*, *Pseudomonas zhaodongensis*, and *Rhizobium yanglingense*. The exclusive species of sample S8 include nine species, identified as *Achromobacter deleyi*, *Achromobacter spanius*, *Bacillus atrophaeus*, *Bacillus kexueae*, *Brachybacterium faecium*, *Erwinia tasmaniensis*, *Patulibacter americanus*, *Pseudomonas atacamensis*, and *Rhodococcus sovatensis* respectively. The exclusive species of sample S9 include one, identified as *Bacillus badius* (Figure 2). Therefore, there are some enormous differences in the endophytic bacteria community structure between the nine samples.

### 3.2. Comparison of Isolation Effects between Different Media

With the purpose of finding an appropriate culture medium for the isolation of endophytic bacteria associated with halophytes growing on the shore of the western Aral Sea Basin in Uzbekistan, we compared and analyzed the isolation effect of 12 different media. After comparison, it was found that the best medium for isolating endophytic bacteria related to halophytes from the western Aral Sea Basin is M7, which takes tryptone and soybean peptone as the carbon-nitrogen source. A certain number of endophytic bacteria were isolated from all nine halophyte samples in both M1 and M7 media. On the contrary, there are no endophytic bacteria isolated from nine halophyte samples using M4 and M8 media, which take L-asparagine and citric acid as the carbon-nitrogen source. Therefore, M4 and M8 media were totally unsuitable for isolating endophytic bacteria related to halophytes from the western Aral Sea (Figure 3).

### 3.3. Comparison of the Isolated Effect of Endophytic Bacteria Isolated from Halophytes between Different Exposed Zones

To understand and explore the isolation effect and the difference of community composition of endophytic bacterial isolates from dominant halophytes growing on the different bared zone, the Upset plot was drawn using the R package “UpSet R”. (Figure 4). A total of 19 genera were detected across six exposed zones with one genus common to all exposed zones, which was identified as *Bacillus*. Ten genera of endophytic bacteria were isolated from the dominant halophytes of the ODZ group. The number of genera exclusive to the ODZ group was five, which were identified as *Achromobacter*, *Brachybacterium*, *Erwinia*, *Patulibacter*, and *Rhodococcus* respectively. Nine genera of endophytic bacteria were isolated from the dominant halophytes of the E1970S group. The number of genera exclusive to the exposed zone E1970S was three, which were identified as *Clavibacter*, *Mycolicibacterium*, and *Rhizobium* respectively. Four genera of endophytic bacteria were isolated from the dominant halophytes of the E1980S group. The number of genera exclusive to the exposed zone E1980S was two, which were identified as *Mesobacillus* and *Novosphingobium* respectively. Six genera of endophytic bacteria were isolated from the dominant halophytes of the E1990S group. The number of genera exclusive to the exposed zone E1990S was two, which were identified as *Brevibacterium* and *Corynebacterium* respectively. Five genera of endophytic bacteria were isolated from the dominant halophytes of the E2009S group. The number of genera exclusive to the exposed zone E2009S was 1, which was identified as *Micrococcus*. Three genera of endophytic bacteria were isolated from the dominant halophytes of the E2000S group. However, there was no one exclusive genus in exposed zone E2000S. The isolated numbers of endophytic bacteria from halophytes in different exposed zones were also displayed in Figure 4. It can be found that the isolated numbers of endophytic bacteria associated with halophytes from the western Aral Sea showed a decreasing trend with the shortening of exposure time. This may be because the shorter the exposure time of the lake bed, the worse the vegetation restoration and coverage so that the endophytic bacteria are less enriched in the plants during a limited time.

### 3.4. Production of Protease, Lipase, and Cellulose by Endophytic Bacterial Isolates

In this experimental part, 42 different top-hit taxa were selected to test enzyme-producing capability. The results of enzyme-producing endophytic bacterial isolates were analyzed and visualized by Microsoft Excel 2019 (Figure 5). The screening results of protease-producing endophytic bacterial isolates indicated that 52% of the tested endophytic bacterial isolates showed negative, only one endophytic bacterial isolate S6-11 (size: 1382 bp) showed strong activity, which was identified as *Nocardiopsis trehalosi* (similarity: 99.78%) (Figure 5A). Similarly, the screening results of lipase-producing endophytic bacterial isolates showed that half of the tested endophytic bacterial isolates showed negative, while only one endophytic bacterial isolate S4-30 (size: 1327 bp) showed strong activity, which was identified as *Streptomyces puniceus* (similarity: 100%) (Figure 5B). Although the screening results of cellulose-producing endophytic bacterial isolates showed that 62% of the tested endophytic bacterial isolates showed negative, five endophytic bacterial isolates S1-21 (size: 1350 bp), S9-9 (size: 1408 bp), S1-24 (size: 1322 bp), S2-14 (size: 1320 bp), and S4-30 (size: 1327 bp) displayed strong activity, which were identified as *Bacillus tequilensis* (similarity: 99.85%), *Nocardiopsis synnemataformans* (similarity: 98.50%), *Streptomyces calvus* (similarity: 100%), *Streptomyces griseoviridis* (similarity: 100%), and *Streptomyces puniceus* (similarity: 100%) respectively (Figure 5C). Based on the comparison of screening results, we found that six endophytic bacterial isolates have all the ability to produce these three tested enzymes (Appendix A).

### 3.5. Antifungal Activity of Endophytic Bacterial Isolates

In this experimental part, the same 42 different top-hit taxa were also selected to test the antagonistic effect of apple canker (*Valsa mali*). The results showed that only eight endophytic bacterial isolates exhibited significant inhibition to the growth of *Valsa mali* (Figure 6). Nevertheless, the average inhibition rate of endophytic bacterial isolate S1-20 (size: 1352 bp), which was identified as *Bacillus siamensis* (similarity: 99.93%), was the highest up to about 73.33%, and compared with other isolates, it has a significant difference (LSD test, *p* < 0.05) (Figure 6A). After that, we screened eight endophytic bacterial isolates with the ability to inhibit against *Valsa mali* to test the other three kinds of pathogenic fungi (*Verticillium dahliae*, *Fusarium oxysporum*, and *Fusarium solani*). Our findings showed that the only endophytic bacterial isolate that exhibited the inhibition effect on *Fusarium oxysporum* and *Fusarium solani* is S1-20. The average inhibition rates of the isolate S1-20 inhibiting *Fusarium oxysporum* and *Fusarium solani* were 68.14% and 51.21%. We obtained four endophytic bacterial isolates: S1-20 (size: 1352 bp), S1-21 (size: 1350 bp), S8-2 (size: 1056 bp), and S8-19 (size: 1324 bp) with the ability to inhibit *Verticillium dahliae* via a screening experiment, which were identified as *Bacillus siamensis* (similarity: 99.93%), *Bacillus tequilensis* (similarity: 99.85%), *Pantoea vagans* (similarity: 99.43%), and *Achromobacter deleyi* (similarity: 100%) respectively. In addition, their average inhibition rates were 34.06%, 34.06%, 16.67%, and 16.67% respectively (Appendix A). Figure 6B and Figure 7 are exhibitions of isolate S1-20, inhibiting four kinds of pathogenic fungi. Finally, we evaluated the difference in inhibition rates of endophytic bacterial isolate S1-20, suppressing four pathogenic fungi. It turns out that the endophytic bacterial isolate S1-20 presented the best inhibition effect and the highest inhibition rate on *Valsa Mali*, followed by *Fusarium oxysporum*, *Fusarium solani*, and *Verticillium dahliae* (Figure 8). The LSD test showed that the inhibition rates of isolate S1-20 against four kinds of pathogenic fungi were marked by different letters. Therefore, there were significant differences in the inhibition rates of isolate S1-20 against four kinds of pathogenic fungi.

## 4. Discussion

### 4.1. Cultivable Endophytic Bacterial Community Composition Associated with the Halophytes Growing on the Shore in the Western Aral Sea Basin

Based on the field investigation of vegetation restoration in the degraded zone of the Western Aral Sea, we found that the species and distribution of plants in the degraded zone of different lakes are quite different (Appendix A). In this study, based on the culture-dependent method, 289 endophytic bacterial isolates affiliated to 3 phyla, 6 classes, 14 orders, 17 families, and 19 genera were isolated from nine halophytic samples of six dominant halophytes growing on the shore of the western Aral Sea of Uzbekistan. The phylum *Firmicutes* accounted for 66% of the total number of isolates, followed by *Actinobacteria* 23%, and *Proteobacteria* 11% (Figure 1A). The predominant genera of endophytic bacteria associated with halophyte from the western Aral Sea were *Bacillus*, *Staphylococcus*, and *Streptomyces*, accounting for 38.5%, 24.7%, and 12.5% of the total number of isolates respectively (Figure 1C). Wang et al. investigated the endophytic bacteria of four *Chenopodiaceae* halophytes (including *Borszczowia aralocaspica* Bge., *Salicornia europaea* L., *Salsola affinis* C. A. Mey, and *Anabasis elatior* (C. A. Mey.) Schischk) in Xinjiang by using two isolation methods and 10 media and found that *Streptomyces* and *Bacillus* were the main dominant groups [31]. When studying the developmental peculiarities of *Chenopodium quinoa*, Pitzschke demonstrated that 100% of quinoa seeds are inhabited by diverse members of the genus *Bacillus*, which are motile, vertical inheritance, and reside in all seedling organs [32]. In our findings, we isolated 32 endophytic bacterial isolates belonging to the genus *Bacillus* from *Chenopodium album* L., which were identified as six different top-hit taxa (Appendix A). The richness of S5 and S6 (*Haloxylon ammodendron* (C. A. Mey.) Bunge) was the lowest in our analysis. According to the search result from PubMed, nowadays, only two new endophytic bacterial species related to *Haloxylon ammodendron* have been published [33,34]. The diversity of S7 was the lowest in our results because only *Bacillus swezeyi* was isolated. Unlike our results, in 2020, Zhang et al. investigated the microbial community composition of *Alhagi sparsifolia* applied 16S rRNA gene amplicon sequencing and found that the dominant class belongs to *Gammaproteobacteria* with relative abundances of 94.8 to 98.2% in all root endosphere samples [35]. Anyway, the endophytic bacteria associated with halophytes have the characteristics of diversity, universality, and specificity [36]. Consequently, in the future, it is necessary to deeply explore endophytic bacterial diversity related to halophytes growing on the shore of the western Aral Sea by combining the culture-dependent method with high throughput sequencing.

### 4.2. The Exploration on the Best Isolation Protocol of Halophyte Endophytic Bacteria

In this paper, the isolation of endophytic bacterial isolates from halophytes was performed using 12 kinds of different media through the spread plate method and streak plate method. According to comparison, it was found that the M7 medium taking yeast extract and tryptone as the carbon source was more effective in the isolation of endophytic bacteria associated with halophytes growing on exposed zones in the western Aral Sea (Figure 3). After Huang et al. investigated root-associated bacteria from *Platycodon grandiflorum*, a medicinal plant commonly growing in East Asia, they found only a few bacterial lineages were found in all cultural media, and most were only enriched and obtained on a single culture medium [37]. Our findings, as well as previous studies, have all shown that multiple media have a greater possibility of obtaining a broader range of bacterial lines. This is an extremely valuable guide to choose multiple media of different nutritional components for the isolation of endophytic bacteria.

### 4.3. The Screening of the Endophytic Bacterial Isolates Producing Enzyme and Inhibiting Pathogenic Fungi

Biological control of plant pathogens using endophytic bacteria is an efficient method and a friendly environment way for inhibiting plant diseases [38]. Endophytic bacteria associated with halophytes are rich sources of secondary metabolites with anti-pathogenic fungi activity, and they spend their whole life cycle living in plant tissues without inducing any infections or diseases [39]. In our study, we analyzed the enzyme-producing activity and the antimicrobial activity of a diverse collection of endophytic bacteria isolated from halophytes, to determine which isolate was provided with the strong enzyme-producing activities and antagonistic activities repressing pathogens. The screening for antagonistic activity was conducted by co-cultivating the endophytic bacteria associated with halophytes with common phytopathogenic fungi of the potato wilt (*Fusarium oxysporum* and *Fusarium solani*), verticillium wilt of cotton (*Verticillium dahliae*), and apple canker (*Valsa mali*) in vitro [40,41,42]. In our assays, several endophytic bacteria displayed antagonistic effects, which were identified as *Bacillus subtilis*, *Bacillus siamensis*, *Bacillus tequilensis*, *Pantoea vagans*, *Bacillus atrophaeus*, *Bacillus kexueae*, *Achromobacter deleyi*, and *Nocardiopsis dassonvillei subsp. dassonvillei* respectively (Appendix A). According to previous research results, it is not difficult to find that the genus *Bacillus* was the dominant genus, with high antimicrobial activity against nearly all pathogenetic fungi, which is consistent with our research results [43]. In our screening findings, only one endophytic bacterial isolate, S1-20, representing *Bacillus siamensis*, possessed the ability to resist four tested kinds of pathogenic fungi and exhibited moderate-high inhibition rates. Besides, we also detected that isolate S1-20 had both protease-producing and cellulose-producing activities (Appendix A). *Bacillus siamensis* was, for the first time, isolated from salted crab in Thailand and phenotypic and chemotaxonomic characteristics, including phylogenetic analyses, showed that the novel isolate was a member of the genus *Bacillus* [44]. As early as 2018, Xu et al. demonstrated the broad-spectrum antibacterial activity of *Bacillus siamensis*, identified cyclic lipopeptides from *Bacillus siamensis* for the first time, and used them to suppress the growth of various multidrug-resistant bacterial pathogens [45]. Based on our research and previous studies, we can conclude that *Bacillus siamensis* has extensive resistance to pathogenic bacteria and fungi. For this reason, *Bacillus siamensis* has a broad development and application prospect in agricultural production. Shurigin et al. [46] reported that antifungal activity of endophytic bacteria such as *Paenibacillus*, *Stenotrophomonas*, *Pseudomonas*, *Brevibacterium,* and *Pantoea* inhibited phytopathogenic fungi *Rhizoctonia solani*, *Fusarium culmorum*, and *Fusarium solani.* Certainly, a further study is necessary for the pot as well as field experiments to verify the antagonistic effect of the isolate S1-20 (*Bacillus siamensis*).

## 5. Conclusions

A total of 289 endophytic bacteria belonging to the phyla *Firmicutes*, *Actinobacteria*, and *Proteobacteria* were isolated from nine halophytes growing on the shore of the western Aral Sea in Uzbekistan. The predominant genera were *Bacillus*, *Staphylococcus*, and *Streptomyces* in endophytic bacterial isolates isolated from nine halophytes. The M7 medium, which took tryptone and soybean peptone as the carbon-nitrogen source, was the best medium for isolating halophyte-related endophyte bacteria in the western Aral Sea region. The isolated number of endophytic bacteria associated with halophytes growing on the western Aral Sea showed a declining trend along with the shortening of exposure time. We acquired one endophytic bacterial isolate with the strong ability to produce protease, one endophytic bacterial isolate with the strong ability to produce lipase, and five endophytic bacterial isolates with the strong ability to produce cellulose. Simultaneously, we also found that six endophyte bacterial isolates have the full capability of producing these three tested enzymes. Eight endophytic bacterial isolates capable of significantly inhibiting the growth of *Valsa mail* were obtained. Moreover, one single isolate S1-20 (*Bacillus siamensis*) also has all the inhibitory effects on *Verticillium dahliae*, *Fusarium oxysporum*, and *Fusarium solani*. Our results suggest that halophytes are sources for the selection of microbes that can protect plant growth under adverse conditions. Besides, further research is needed for the glasshouse as well as field experiments to verify the biological control efficacy of selected endophytic bacteria.

## Figures and Tables

**Figure 1 microorganisms-09-01448-f001:**
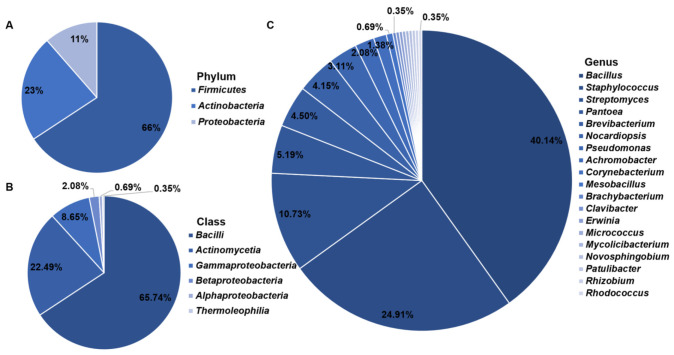
The percentage of endophytic bacterial isolates isolated from nine halophytic samples in the western Aral Sea at the phylum (**A**), class (**B**), and genus (**C**) level.

**Figure 2 microorganisms-09-01448-f002:**
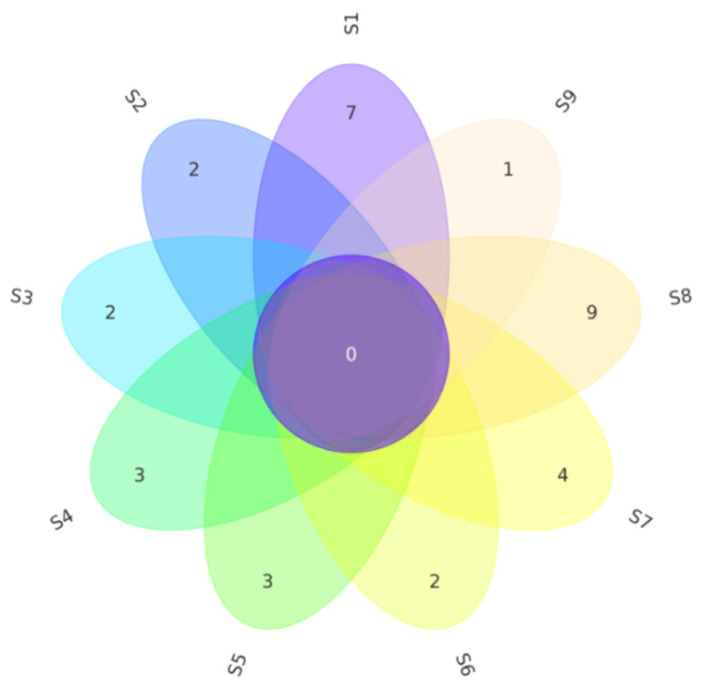
Flower diagram shows the difference of endophytic bacteria among different samples based on culture-dependent methods.

**Figure 3 microorganisms-09-01448-f003:**
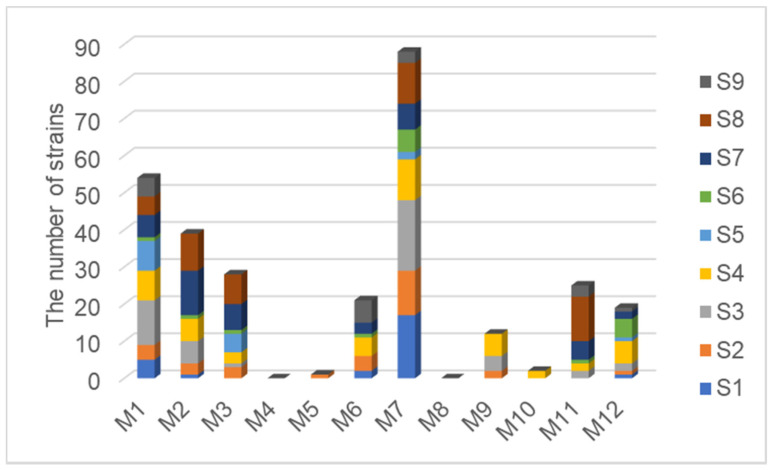
Comparison of endophytic bacteria isolation by different media.

**Figure 4 microorganisms-09-01448-f004:**
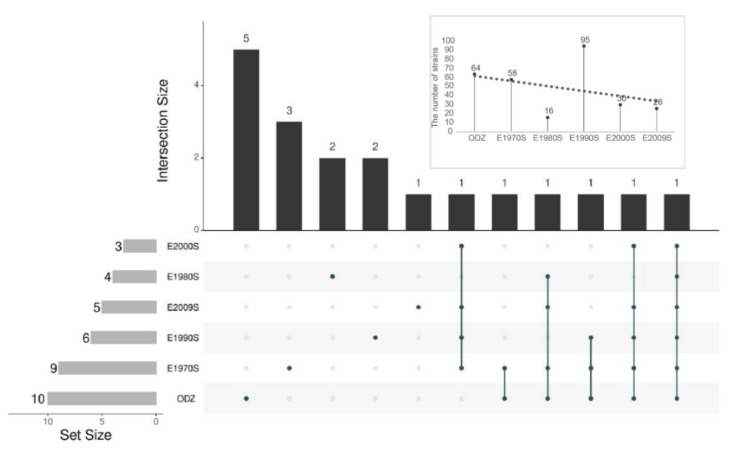
The Upset plot of endophytic bacteria from nine halophytes at the genus level.

**Figure 5 microorganisms-09-01448-f005:**
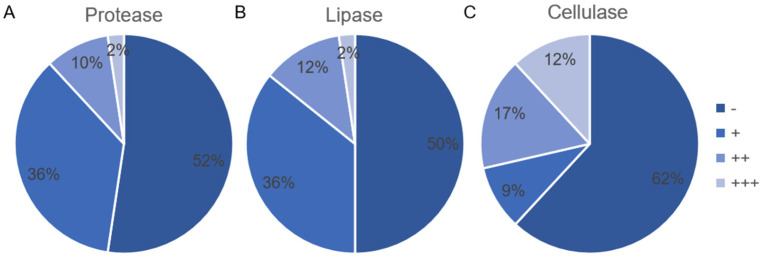
Screening results of endophytic bacteria isolates producing protease (**A**), lipase (**B**), and cellulose (**C**) (Notes: -: negative (E = 1), +: weak (1 < E ≤ 2), ++: moderate (2 < E ≤ 3), +++: strong (E > 3)).

**Figure 6 microorganisms-09-01448-f006:**
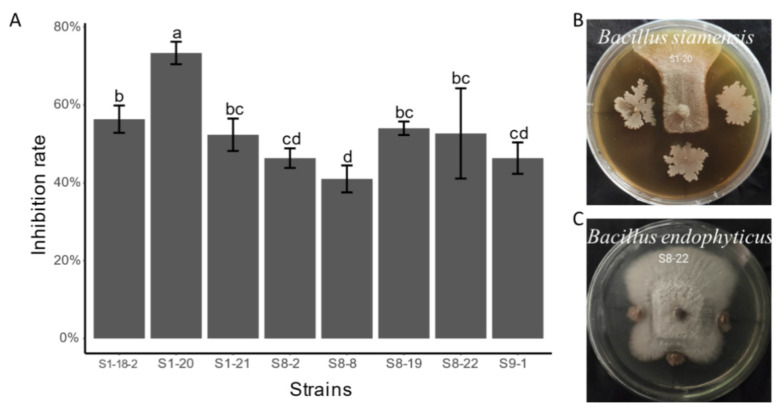
Screening results of endophytic bacterial isolates against pathogenic fungi of apple canker (*Valsa mali*), (**A**): bar chart of inhibition rate of endophytic bacteria against *Valsa mali*, (**B**): antagonistic effect picture of isolate S1-20, (**C**): antagonistic effect picture of isolate S8-22. The LSD test was used to check whether there was a significant difference in the inhibition rate among the different endophytic bacterial isolates. There were significant differences in the inhibition rate of endophytic bacterial isolates labeled with different letters. There was no significant difference in the inhibition rate between endophytic bacterial isolates labeled with repeated letters.

**Figure 7 microorganisms-09-01448-f007:**
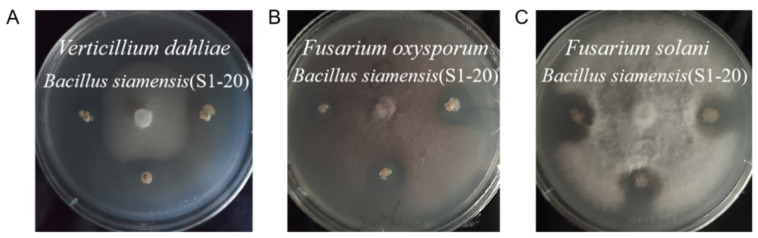
Exhibition of antagonistic effects of endophytic bacterial isolate S1-20 against pathogenic fungi ((**A**): *Verticillium dahliae*, (**B**): *Fusarium oxysporum*, and (**C**): *Fusarium solani*).

**Figure 8 microorganisms-09-01448-f008:**
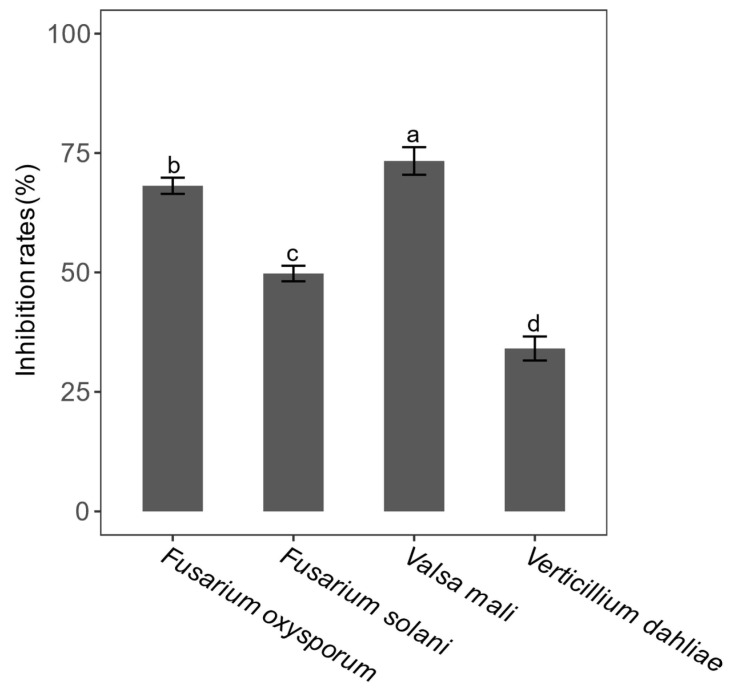
Inhibition rates of endophytic bacteria isolate S1-20 against four different pathogenic fungi (*Valsa mali*, *Verticillium dahliae*, *Fusarium oxysporum*, and *Fusarium solani*). The LSD test was used to detect whether there were significant differences in the inhibition rates of endophytic bacteria isolate S1-20 against four kinds of pathogenic fungi. There were significant differences in the inhibition rate of endophytic bacterial isolates labeled with different letters.

**Table 1 microorganisms-09-01448-t001:** Samples information.

Sample Num.	Sample Name	Exposed Zone	The Distance to the Shoreline (m)	Location
S1	*Chenopodium album* L.	E2009S	~200	44°29′50.42″ N~44°29′50.4233″ N, 58°12′43.80″ E~58°13′42.31″ E
S2	*Chenopodium album* L.	E2000S	~300	
S3	*Chenopodium album* L.	E1990S	~700	
S4	*Salsola collina* Pall.	E1990S	~700	
S5	*Haloxylon ammodendron* (C. A. Mey.) Bunge	E1980S	~1000	
S6	*Haloxylon ammodendron* (C. A. Mey.) Bunge	E1970S	~1200	
S7	*Alhagi sparsifolia* Shap.	E1970S	~1200	
S8	*Anabasis eriopoda* (Schrenk) Benth. ex Volkens	ODZ	>1500	
S9	*Anabasis truncata* (Schrenk) Bunge	ODZ	>1500	

**Table 2 microorganisms-09-01448-t002:** Medium component.

Medium Num.	Medium Component (1 L), Agar 15 g, pH 7.2
M1	Yeast extract 0.25 g, K_2_HPO_4_ 0.5 g, NaCl 30 g
M2	Na_2_C_2_O_4_ 2 g, Casamino acids 0.5 g, KH_2_PO_4_ 0.3 g, Na_2_HPO_4_·12H_2_O 0.5 g, NaCl 30 g, ZnSO_4_·7H_2_O 0.02 g, CaCl_2_ 0.5 g
M3	Glycerol 10 g, L-Asparagine 1 g, NaCl 30 g, K_2_HPO_4_ 1 g, FeSO_4_·7H_2_O 0.001 g, MnSO_4_·H_2_O 0.001 g, ZnSO_4_·7H_2_O 0.001 g, CuSO_4_·5H_2_O 0.001 g
M4	Na_2_SeO_3_ 1 g, L-Asparagine 1 g, MgSO_4_·7H_2_O 0.5 g, FeSO_4_·7H_2_O 0.05 g, NaCl 30 g, CaCl_2_ 0.5 g, KH_2_PO_4_ 0.2 g,
M5	Sodium propionate 2 g, L-Asparagine 1 g, (NH_4_)_2_SO_4_ 0.1 g, KCl 0.1 g, MgSO_4_·7H_2_O 30 g, FeSO_4_·7H_2_O 0.05 g
M6	Chitin 2 g, NaCl 30 g, K_2_HPO_4_ 0.7 g, KH_2_PO_4_ 0.3 g, MgSO_4_·7H_2_O 0.5 g, MnCl_2_ 0.001 g
M7	Tryptone 15 g, Soybean peptone 5 g, NaCl 30 g
M8	Citric Acid 0.12 g, Ferric citrate 0.12 g, NaCl 30 g, NaNO_3_ 0.5 g, K_2_HPO_4_ 0.4 g, MgSO_4_·7H_2_O 0.2 g, CaCl_2_ 0.5 g
M9	Cellulose microcrystalline 2.5 g, Proline 1 g, NaCl 30 g, KNO_3_ 0.25 g, MgSO_4_·7H_2_O 0.2 g, K_2_HPO_4_ 0.2 g, CaCl_2_ 0.5 g, FeSO_4_·7H_2_O 0.01 g
M10	Sodium propionate 2 g, Arginine 1 g, NaCl 30 g, MgSO_4_·7H_2_O 1 g, KH_2_PO_4_ 0.1 g, FeSO_4_·7H_2_O 0.05 g
M11	Starch 20 g, KNO_3_ 1 g, NaCl 0.5 g, MgSO_4_·7H_2_O 0.5 g, K_2_HPO_4_ 0.5 g, FeSO_4_·7H_2_O 0.01 g
M12	Sodium succinic hexahydrate 1 g, NaCl 30 g, L-Asparagine 1 g, KH_2_PO_4_ 0.9 g, K_2_HPO_4_ 0.5 g, KCl 0.3 g, CaCl_2_ 0.2 g, MgSO_4_·7H_2_O 0.1 g, FeSO_4_·7H_2_O 0.001 g

**Table 3 microorganisms-09-01448-t003:** Diversity comparison of endophytic bacteria isolated from different samples.

Plant	S1	S2	S3	S4	S5	S6	S7	S8	S9
Phylum	2	2	3	2	3	2	3	3	3
Class	2	2	3	2	3	2	4	5	3
Order	4	2	3	4	3	3	7	9	5
Family	5	3	4	5	3	4	7	9	5
Genus	5	3	4	5	4	4	7	10	5
Isolates	26	30	46	49	16	16	42	46	18
Richness	13.00	7.00	7.00	5.00	5.00	5.00	8.00	16.00	7.00
Shannon	2.08	1.44	1.13	1.40	1.24	1.19	0.85	2.52	1.61
Simpson	0.79	0.68	0.59	0.73	0.63	0.61	0.34	0.90	0.75
Pielou	0.81	0.74	0.58	0.87	0.77	0.74	0.41	0.91	0.83

## Data Availability

Raw sequence data reported in this paper have been deposited in the GenBank in the NCBI under accession number MW664036–MW664324.

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
