# Peer review of "Diversity and Biocontrol Potential of Cultivable Endophytic Bacteria Associated with Halophytes from the West Aral Sea Basin"

_microorganisms, 2021, doi:10.3390/microorganisms9071448_

Round 1

Reviewer 1 Report

The manuscript is prepared on an interesting topic, but contains shortcomings that need to be eliminated. In its current form, it is not suitable for publishing.

Material and methodology - there is no fundamental justification for the selection of 6 botanical species and their different representation. When using different species, the results are expected and difficult to interpret. This information is crucial to the concept and goals of the entire manuscript. It is not possible to accept a manuscript for publication without sufficient explanation.

The primer combination was suggested by the authors or is taken over. If it is taken over, it is necessary to state the citation similarly to the PCR protocol. Was one or both primers used for sequencing?

The results (text) indicate the size of PCR products, resp. their range. This is shown in TableS1 and is highly variable (650 - 1,400 bp). Why is this not described, resp. discussed?

It is necessary to unify the writing of genera and species in italics throughout the text of the manuscript.

Bar graphs (e.g. Figures 6 and 7) show a statistical designation that is not explained. The description of the pictures and the table must be self-explanatory!

The manuscript also includes 3 xls files. Table S1 (xls) has several sheets. There are no names of tables, or explanations of the content (sometimes also color resolution), or unify the format of numbers and text. Similarly, the headings for Table S2 and S3 are missing! Need to add!

In the References section, inconsistent processing of references, eg names of authors somewhere in full, elsewhere with initials, etc. Nunto process according to the instructions for authors.

I recommend the manuscript for publication after major revision, with the fact that it is necessary to add essential information within the material and methodology, which should be reflected in the discussion of the manuscript. Without these changes, it is not possible to accept the manuscript.

Author Response

Thank you for your letter and for the reviewer’s comments concerning our manuscript entitled “Diversity of cultivable endophytic bacteria associated with halophytes from the west Aral Sea Basin with biocontrol potential” (Manuscript ID: microorganisms-1267002). These comments are all valuable and very helpful for revising and improving our paper, as well as the important guiding significance to our researches. We have studied comments carefully and have made correction which we hope meet with approval. Revised portion are marked in red in the paper. The main corrections in the paper and the responds to the reviewer’s comments are as flowing:

  1. Material and methodology - there is no fundamental justification for the selection of 6 botanical species and their different representation. When using different species, the results are expected and difficult to interpret. This information is crucial to the concept and goals of the entire manuscript. It is not possible to accept a manuscript for publication without sufficient explanation.

Response: Thank you for your suggestion. The relative contents have been added in 2.1. Sample collection and site description. “These nine halophyte samples belong to six halophytic species, which are widely distributed on the shore of the Western Aral Sea. This indicated that these halophytes are the dominant species in the local area. Therefore, these dominant plants grown on the shore of the Western Aral Sea were selected to investigate endophytic bacterial diversity and community structure.”

  1. The primer combination was suggested by the authors or is taken over. If it is taken over, it is necessary to state the citation similarly to the PCR protocol. Was one or both primers used for sequencing?

Response: Thank you for your suggestion and question. The relative reference has been cited in our new manuscript. Paired-end primers (27F and 1492R) were used for sequencing.

  1. The results (text) indicate the size of PCR products, resp. their range. This is shown in TableS1 and is highly variable (650 - 1,400 bp). Why is this not described, resp. discussed?

Response: Thank you for your suggestion and question. The length and similarity of 16S rRNA gene sequences of endophytic bacteria strains with enzyme-producing and antifungal activities were marked in our new manuscript. Although the length of the 16S rRNA gene sequences of a few endophytic bacterial strains is short, it can be found that their similarities are high, which can basically determine their genus and species information.

  1. It is necessary to unify the writing of genera and species in italics throughout the text of the manuscript.

Response: Thank you for your suggestion. It has been revised.

  1. Bar graphs (e.g. Figures 6 and 7) show a statistical designation that is not explained. The description of the pictures and the table must be self-explanatory!

Response: Thank you for your suggestion. The description of relevant significant markers has been supplemented in our new manuscript and the titles of Figures.

  1. The manuscript also includes 3 xls files. Table S1 (xls) has several sheets. There are no names of tables, or explanations of the content (sometimes also color resolution), or unify the format of numbers and text. Similarly, the headings for Table S2 and S3 are missing! Need to add!

Response: Thank you for your suggestion. It has been revised.

  1. In the References section, inconsistent processing of references, eg names of authors somewhere in full, elsewhere with initials, etc. Nunto process according to the instructions for authors.

Response: Thank you for your suggestion. It has been revised.

  1. I recommend the manuscript for publication after major revision, with the fact that it is necessary to add essential information within the material and methodology, which should be reflected in the discussion of the manuscript. Without these changes, it is not possible to accept the manuscript.

Response: Thank you for your suggestion and recommendation. Some essential informations have been added in our new manuscript.

Reviewer 2 Report

I have reviewed the manuscript ‘Diversity of cultivable endophytic bacteria associated with halophytes from the west Aral Sea Basin with biocontrol potential’ by Gao et al. and have the following comments:

The manuscript covers the isolation and characterization of 289 endophytic bacteria from halophytes using a number of isolation media and have further tested them all for enzyme activity and the ability to inhibit (not fighting against!) plant pathogenic fungi. The experimental work is rather comprehensive and the authors have done a fairly comprehensive analysis and I commend them for their excellent study.

Nevertheless I would like to suggest a few changes to improve the manuscript.

The pie charts in figure 1 are difficult to read and could be replaced by a better representation of the data.

Some sections are explained in too much detail  as in lines 185-206 when the data is shown in Table 3. However they have to explain their methodology and results on the Upset plot of the isolates from the ODZ as shown in Table 4, in a more understandable  manner. I could not understand the purpose of this analysis.

The potential biocontrol activity was assessed against pathogenic fungi using an in vitro plate test with a fairly elaborate analysis of the zone of inhibition. While it is as good a measure as any it must be remembered that the size of t h e zone of inhibition is determined by  the amount, potency and diffusibility of the antifungal compound produced and therefore the size of the zone may not convey a true comparison of the value of the activity noted. Similarly there is a poor correlation between in vitro and in planta activity, which argues for glasshouse (not potty!) evaluation of all strains.

Lastly in addition to ‘fighting’ and ‘potty’ there other terms such as ‘heavily’, ‘vigorously’ and ‘powerfully’ that are rarely used as adverbs in the scientific literature. It would be advisable to have an editor proof the final version.

Author Response

Thank you for your letter and for the reviewer’s comments concerning our manuscript entitled “Diversity of cultivable endophytic bacteria associated with halophytes from the west Aral Sea Basin with biocontrol potential” (Manuscript ID: microorganisms-1267002). These comments are all valuable and very helpful for revising and improving our paper, as well as the important guiding significance to our researches. We have studied comments carefully and have made correction which we hope meet with approval. Revised portion are marked in red in the paper. The main corrections in the paper and the responds to the reviewer’s comments are as flowing:

  1. The pie charts in figure 1 are difficult to read and could be replaced by a better representation of the data.

Response: Thank you for your suggestion. We have made further adjustments to the pie charts, such as enlarging the font, raising the definition and so on. Futhermore, we replace the pie chart at the order level with the pie chart at the class level which includes fewer assigned taxa to simplify the pie plot.

  1. Some sections are explained in too much detail  as in lines 185-206 when the data is shown in Table 3. However they have to explain their methodology and results on the Upset plot of the isolates from the ODZ as shown in Table 4, in a more understandable  manner. I could not understand the purpose of this analysis.

Response: Thank you for your suggestion. A more detailed description of the subset plot has been added to our new manuscript. The subset plot is a way to show the common populations among the different sample groups and exclusive populations within a sample group to understand and explore the isolation effect and the difference of community composition of endophytic bacterial strains from dominant halophytes grown on the different bared zone in our study.

  1. The potential biocontrol activity was assessed against pathogenic fungi using an in vitro plate test with a fairly elaborate analysis of the zone of inhibition. While it is as good a measure as any it must be remembered that the size of the zone of inhibition is determined by  the amount, potency and diffusibility of the antifungal compound produced and therefore the size of the zone may not convey a true comparison of the value of the activity noted. Similarly there is a poor correlation between in vitro and in planta activity, which argues for glasshouse (not potty!) evaluation of all strains.

Response: Thank you for your sharing. We agree with your opinion. In vitro plate confrontation experiment is only the first step of screening endophytic bacterial strains inhibiting pathogenic fungi. Further field experiments are needed to obtain functional strains with resistance to plant pathogenic fungi. Moreover, these follow-up experiments (such as glasshouse and field experiments) are also in our next research plan.

  1. Lastly in addition to ‘fighting’ and ‘potty’ there other terms such as ‘heavily’, ‘vigorously’ and ‘powerfully’ that are rarely used as adverbs in the scientific literature. It would be advisable to have an editor proof the final version.

Response: Thank you for your suggestion. It has been revised.

We tried our best to improve the manuscript and made some changes in the manuscript. These changes will not influence the content and framework of the paper. And here we did not list the changes but marked in red in revised paper.

We appreciate for Editors and Reviewer’s warm work earnestly, and hope that the correction will meet with approval.

Once again, thank you very much for your comments and suggestions!

Round 2

Reviewer 1 Report

The authors accepted most of the comments. Nevertheless, the manuscript contains errors of a formal nature that need to be corrected:
- Table S2 - similarity (%) - wrong number format (decimal point is correct); the table has 13 sheets - where the names of species and genera (italics) are also incorrectly stated.
- Table S4 - wrong number format (decimal point is correct).
In general, it is necessary to unify the writing in italics of Latin names (Figure 1, 8, etc.)
References - journal names are not uniformly and correctly cited (see Instruction for Authors)
I recommend the manuscript for publication after minor corrections that are of a formal nature.